# Comparison of Single and Combined Use of Ergothioneine, Ferulic Acid, and Glutathione as Antioxidants for the Prevention of Ultraviolet B Radiation-Induced Photoaging Damage in Human Skin Fibroblasts

**Gregory J. Tsay [1,2,†], Shin-Yi Lin [3,†], Chien-Yu Li [4], Jeng-Leun Mau [5] and Shu-Yao Tsai [3,6,*]**

[1] Division of Immunology and Rheumatology, Department of Internal Medicine, China Medical University Hospital, Taichung 40447, Taiwan; jjtsay@mail.cmu.edu.tw

[2] Department of Internal Medicine, School of Medicine, China Medical University Hospital, China Medical University, Taichung 40447, Taiwan

[3] Department of Food Nutrition and Health Biotechnology, Asia University, Wufeng, Taichung 41354, Taiwan; b9136054@hotmail.com

[4] Department of Neurosurgery, Asia University Hospital, Taichung 41354, Taiwan; sbrain.lee@mail.cmu.edu.tw

[5] Department of Food Science and Biotechnology, National Chung Hsing University, Taichung 40227, Taiwan; jlmau@dragon.nchu.edu.tw

[6] Department of Biotechnology, National Formosa University, 64 Wunhua Rd., Huwei Township 632301, Yunlin County, Taiwan

* Correspondence: sytsai@asia.edu.tw or sytsai439@gmail.com

† These authors contributed equally to this work.

**Abstract:** Ultraviolet B (UVB) irradiation can cause human skin damage or skin aging and wrinkle formation through photochemical reactions. Antioxidative substances may ameliorate UV damage. In this study, the anti-photoaging activity of three antioxidants—ergothioneine, ferulic acid, and glutathione—was investigated after UVB irradiation of Hs68 human skin fibroblast cells. The cells treated with these three antioxidants appeared similar to unirradiated control cells. UVB irradiation decreased cell viability by 26% compared to that of unirradiated control cells. However, the addition of either single or combined antioxidants enhanced cell viability after UVB irradiation. These three antioxidants can inhibit the production of reactive oxygen species (ROS) induced by the UVB irradiation of the Hs68 cells. Ergothioneine showed a greater inhibitory effect on matrix metalloproteinase-1 (MMP-1) performance than the other two antioxidants. IL-1 alpha was not detected in the Hs68 cells after exposure to a radiation dose of 150 mJ/cm$^2$. Ergothioneine showed better restoration of type 1 procollagen than either ferulic acid or glutathione. Based on these results, the addition of two antioxidants was expected to restore type I procollagen production. In summary, these results demonstrate that the three tested antioxidants protect the skin against UVB-induced damage. The single and combined use of ergothioneine, ferulic acid, and glutathione has the potential for development as anti-photoaging materials in cosmetic applications.

**Keywords:** antioxidant; ergothioneine; ferulic acid; glutathione; photoaging; human skin fibroblasts

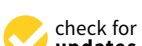

## 1. Introduction

The skin is an important barrier that protects the body from environmental factors, such as sunlight, pollutants, and air. Long-term or large-scale exposure to these environmental factors can cause skin damage, skin cell death, skin aging, and skin cancer [1,2]. Among these factors, ultraviolet (UV) radiation is the main environmental factor that causes erythema, inflammation, photoaging, and skin cancer [3]. The use of sunscreen has been extensively promoted to reduce UV-induced skin damage. Phyto-constituents have been suggested as potential sources for sunscreens due to their UVB absorption capacity and strong antioxidant properties [4–8]. Therefore, antioxidants with absorption spectra

in the UVB range have received considerable attention in the prevention of skin damage caused by UVB.

Commonly used whitening agents such as hydroquinone, kojic acid, arbutin, and azelaic acid can reduce the signs of skin aging; however, there are also reports of their lack of efficacy and potential side effects [9]. Therefore, the necessity to develop new methods that prevent skin aging has become important. It is known that natural compounds are activators of antioxidant genes and may play a role in protecting against UV-induced cytotoxicity [10].

Ergothioneine (ERG, 2-mercaptohistidine trimethylbetaine) is a naturally occurring amino acid that is a thiourea derivative of histidine and contains a sulfur atom on the imidazole ring and trimethylated amine [11]. Many studies have reported ERG to be a potential ingredient in anti-photoaging cosmetic products for its ability to inhibit the expression of the MMP-1 protein and for its anti-inflammatory properties in dermal fibroblasts upon UV irradiation [12]. The human body cannot produce ergothioneine on its own, so it needs to be taken from the diet. ERG has been shown to protect cells from UV-induced ROS production, is more powerful than CoQ10 in preventing lipid peroxidation, and is better than idebenone in directly scavenging peroxide [13]. ERG not only prevents oxidative damage, but may also enable DNA repair after the UV irradiation of cells [14,15]. Glutathione (GSH, $\gamma$-L-Glutamyl-L-cysteinyl-glycine) is a health dietary supplement used in several countries, including the Philippines, Malaysia, Japan, Taiwan, and Thailand [16]. Some studies have shown that glutathione is involved in melanogenesis via mechanisms associated with pheomelanin synthesis, which allows glutathione to act as a whitening agent in certain cosmetic products [17]. Ferulic acid (FA, 4-hydroxy-3-methoxcinnamic acid) is a phenolic compound found in cereal brans, popcorn, artichokes, and some vegetables. FA has shown antidiabetic, antioxidant, antimicrobial, anti-wrinkle, and whitening properties [18–21]. An FA treatment group demonstrated the inhibition of UVB-induced cytotoxicity and apoptosis [22]. In addition, FA has been used as a photoprotective agent in many skin products such as lotions and sunscreens. In vitro and in vivo experiments may be successfully employing FA as topical protective agents against UV radiation-induced skin damage [23]. L-ascorbic acid and $\alpha$-tocopherol are able to stabilize FA, thus endowing cells with dual protection against photoaging and skin cancer [24,25].

UV radiation is divided into three types: UVA, UVB, and UVC. UVB radiation is able to cross the epidermal layer and penetrate the upper layer of the dermis in human skin, causing photochemical cellular damage. UVB is the predominant cause of photoaging in the dermal fibroblast; it causes damage and is associated with reactive oxygen species (ROS) and matrix metalloproteinases (MMPs) [26]. ROS act as messengers in signaling pathways and activate complex signaling molecules such as activator protein-1 (AP-1) and nuclear factor-*k*B (NF-*k*B) [27]. Increased AP-1 activity inhibits type I procollagen synthesis by blocking the transforming growth factorβ1 (TGF-β1) function [28]. In addition, MMP-1 degrades type I procollagen upon UV irradiation [29]. Type I procollagen is a precursor of type I collagen, an important protein synthesized by dermal fibroblasts in the skin [30]. UV radiation causes certain inflammatory reactions in skin cells, including the increased expression of the inflammatory cytokines IL-1$\alpha$, IL-6, and TNF-alpha. With regard to skin photoaging in the dermis, IL-1$\alpha$ may play an important role in collagen synthesis and degradation [31]. Moreover, IL-1$\alpha$ stimulates fibroblasts in the dermis to produce glycosaminoglycans, particularly hyaluronic acid [32].

Although several studies have shown the photo-protective roles of ergothioneine, ferulic acid, and glutathione, there are little or no data demonstrating their in vitro photoprotective potential when combined. The results of a literature review on this topic revealed that, for the past few decades, there have been many studies on UVB photoaging; however, only a few of these studies address the anti-UVB role of ERG, FA, and GSH. The purpose of this study was to evaluate the healing effects of these three antioxidants and their combination in the UVB irradiation of human skin fibroblasts.

## 2. Materials and Methods

### 2.1. Preparation of the Three Antioxidants

Ergothioneine (ERG), ferulic acid (FA), and glutathione (GSH) were purchased from Sigma-Aldrich (St. Louis, MO, USA). The stock concentration of ERG, FA, and GSH was 100 mM. The respective stock was added to deionized water or DMSO to obtain dilutions for each working concentration. The test concentrations of ERG were 1, 10, and 50 μM; FA and GSH test concentrations were 10, 50, and 100 μM. The following concentrations were used in the mixtures: concentrations of ERG at 1, 10, and 50 μM were combined with FA at 10, 50, and 100 μM (E1 + F10, E10 + F50, and E50 + F100); concentrations of FA at 10, 50, and 100 μM were combined with GA at 10, 50, and 100 μM (F10 + G10, F50 + F50, and E100 + G100); and concentrations of ERG at 1, 10, and 50 μM were combined with GA at 10, 50, and100 μM (E1 + G10, E10 + G50, and E50 + G100). The three-antioxidant mixed concentrations combined ERG at 1, 10, and 50 μM with FA and GSH at 10, 50, and 100 μM (E1 + F10 + G10, E10 + F50 + G50, and E50 + F100 + G100) (Table 1).

**Table 1.** List of sample, concentration, and abbreviation.

| Sample | Concentration (μM) | Abbreviation |
|---|---|---|
| Ergothioneine | 1 and 50 | E1, E10 and E50 |
| Ferulic acid | 10, 50 and 100 | F10, F50, and F100 |
| Glutathione | 10, 50 and 100 | G10, G50, and G100 |
| Ergothioneine + Ferulic acid | 1 and 50 + 10, 50 and 100 | E1, E10 and E50 + F10, F50, and F100 |
| Ferulic acid + Glutathione | 10, 50 and 100 + 10, 50 and 100 | F10, F50, and F100 + G10, G50, and G100 |
| Ergothioneine + Glutathione | 1 and 50 + 10, 50 and 100 | E1, E10 and E50 + G10, G50, and G100 |
| Ergothioneine + Ferulic acid + Glutathione | 1 and 50 + 10, 50 and 100 + 10, 50 and 100 | E1, E10 and E50 + F10, F50, and F100 + G10, G50, and G100 |

### 2.2. Cell Culture

Human skin fibroblasts (Hs68, BCRC 60038) were purchased from the Bioresource Collection and Research Center (Hsinchu, Taiwan). Cell culture media were purchased from Gibco (Grand Island, NY, USA). Cells were seeded in Dulbecco's modified Eagle's medium (DMEM) containing 10% fetal bovine serum, L-glutamine (4 mM), and sodium bicarbonate (1.5 g/L), and maintained in a humidity incubator at 37 °C with 5% $CO_2$.

### 2.3. UVB Irradiation

The UVB irradiation equipment was purchased from Tsang-Shin Co. (Taichung, Taiwan). The UVB source, G15T8E Type (Sankyo Denki Co., Taipei, Taiwan), was used to deliver an energy spectrum of UVB radiation (280–360 nm; peak intensity, 310 nm). The height of the lamp was adjusted for the sample to be 15–35 cm away from the light. The radiation energy was detected by the UVX Radiometer UVX 312 (Vilber Lourmat, Collégien, France) [33]. Upon reaching 80% confluency, Hs68 cells were washed three times with phosphate-buffered saline (PBS) after the culture medium was removed. Next, the Hs68 cells in PBS were exposed to UVB radiation at different doses: 0, 100, 150, and 200 mJ/cm$^2$; the irradiation time was 10 min. All irradiation was performed under a thin layer of PBS (0.125 mL per well) in a 96-well plate with a cover. After UVB irradiation, fibroblasts were treated with different concentrations of the single antioxidant or the antioxidant combination in DMEM (containing 1% fetal bovine serum) for 24 h or 48 h, respectively.

### 2.4. Cell Viability Assay

The Hs68 cells were placed at a density of $1 \times 10^5$ cells/mL in 96-well plates for an evaluation of cell viability. After being incubated in a 5% $CO_2$ incubator at 37 °C for 24 h or 48 h, the medium was removed and an MTT (3-(4,5-dimethylthiazol-2-yl)-2,5-diphenyltetrazolium bromide) reagent (5 mg/mL, Sigma-Aldrich, St. Louis, MO, USA) was added to each well and incubated for 2 h. After discarding the supernatant, the insoluble formazan product was dissolved in dimethyl sulfoxide (DMSO, Sigma-Aldrich) for 20 min. Cell viability was measured at 540 nm using a FLUOstar galaxy Microplate Reader (BMG Labtech, Ortenberg, Germany).

### 2.5. Intracellular Reactive Oxygen Species (ROS) Concentration

Hs68 cells were seeded into black 96-well Immuno-plates (Thermo Fisher Scientific, Taipei, Taiwan) at a density of $1 \times 10^5$ cells/mL for 24 h and then exposed to UVB radiation at 150 mJ/cm$^2$. Next, various concentrations of a single antioxidant or a combination of antioxidants in 1% serum-DMEM was incubated with cells at 37 °C for 3 h in the presence of 10 µM of a 2,7-dichlorofluorescein-diacetate reagent (Sigma-Aldrich). Next, DMEM (containing 1% fetal bovine serum) was removed, and the cells were washed twice with 0.125 mL PBS. The cells were subsequently covered with 0.125 mL PBS. The FLUOstar galaxy microplate reader (emission 485 nm, excitation 520 nm) was used measure the fluorescence intensity of the samples. The inhibition of ROS was calculated as follows: Inhibition (%) = $[(\text{Flu}_{ex/em} \ (\text{UVB}) - \text{Flu}_{ex/em} \ (\text{sample})]/[(\text{Flu}_{ex/em} \ (\text{UVB}) - \text{Flu}_{ex/em} \ (\text{control})] \times 100\%$.

### 2.6. Measurement of IL-1 Alpha, Total Matrix Metalloproteinase-1 (MMP-1), and Type 1 Procollagen

Hs68 cells were seeded into 96-well plates at a density of $1 \times 10^5$ cells/mL, cultured for 24 h and then exposed to UVB radiation. Next, various concentrations of a single antioxidant or a combination of antioxidants prepared in 1% serum-DMEM were incubated with UVB-irradiated cells at 37 °C for 24 h. The culture supernatants were collected and analyzed for IL-1 alpha, total matrix metalloproteinase-1 (MMP-1), and procollagen 1 alpha 1 levels with a FLUOstar galaxy Microplate Reader and a DuoSet ELISA Development kit (DY-200, DY901B, and DY6220-05, R&D Systems, Minneapolis, MN, USA). The inhibition of MMP-1 was calculated as follows: Inhibition (%) = $[(\text{Flu}_{ex/em} \ (\text{UVB}) - \text{Flu}_{ex/em} \ (\text{sample})]/[(\text{Flu}_{ex/em} \ (\text{UVB}) - \text{Flu}_{ex/em} \ (\text{control})] \times 100\%$. The recovery of Type 1 procollagen production was calculated as follows: Recovery (%) = $[(\text{Flu}_{ex/em} \ (\text{UVB}) - \text{Flu}_{ex/em} \ (\text{sample})]/[(\text{Flu}_{ex/em} \ (\text{UVB}) - \text{Flu}_{ex/em} \ (\text{control})] \times 100\%$.

### 2.7. Statistical Analysis

SAS software (version 9.4, SAS Institute, Cary, NC, USA) and Sigma Plot 2015 software (version 12.5, SPSS, Chicago, IL, USA) were used for data regression and graphical analysis, respectively. Each subculture was considered a replicate, and at least three subcultures were repeated for each determination (number of independent experiments, $n = 4$–8). The experimental data was expressed as the mean $\pm$ standard deviation (SD), and a one-way classification design was used to perform an analysis of variance to determine Fisher's smallest significant difference at the level of $p > 0.05$.

## 3. Results and Discussion

### 3.1. Cell Viability

MTT assay is a colorimetric method used to evaluate cytotoxic effects. Hs68 skin fibroblast viability with or without UVB radiation exposure was measured using an MTT assay. The effects of the three antioxidants—ERG, FA, and GSH—on the viability of unirradiated Hs68 cells for 24 or 48 h were assessed. In addition, the effects of a single antioxidant, or those of two or three mixed antioxidants on cell viability were investigated. Likewise, Figure 1 displays the ability of the three various antioxidants to prevent UVB radiation-induced photoaging in human skin fibroblasts.

Considering the addition of a single antioxidant, the survival results of skin fibroblasts indicated that ERG (E, 1, 10, and 50 µM), FA (F, 10, 50, and 100 µM), and GSH (G, 10, 50, and 100 µM) did not induce cytotoxic effects in unirradiated Hs68 cells after 24 h (Figure 2); cell viability ranged from 96.29% to 100.67%. Studies have shown that FA has no cytotoxicity at a concentration of 20 µM [21]. Cell viability was decreased from 100% to 95.39% after 48 h. When unirradiated Hs68 cells were treated with high concentrations of FA for 48 h, cell viability was affected. For the F10, F50, and G100 groups (Figure 2A), cell viability significantly differed from that of the controls after 48 h. Considering the two mixed antioxidants, all antioxidants were divided into nine groups and compared with controls

(Figure 2B). Cell viability of the E10 + F50 group was 94.25%; for the other eight groups, cell viability did not significantly differ from that of the unirradiated control cells. Considering the three mixed antioxidants, all antioxidants were divided into three groups (Figure 2C). Cell viability (100.25–102.20%) did not significantly differ from that of unirradiated control cells (100.88%) after 24 h; however, cell viability significantly differed after 48 h.

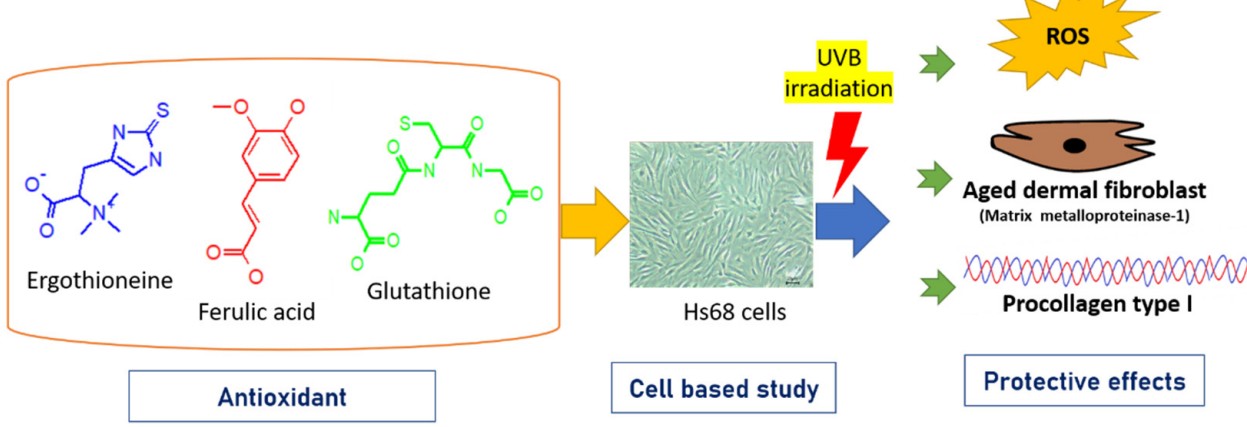

**Figure 1.** Schematic illustration of three types of antioxidants in the prevention of UVB radiation-induced photoaging in human skin fibroblasts.

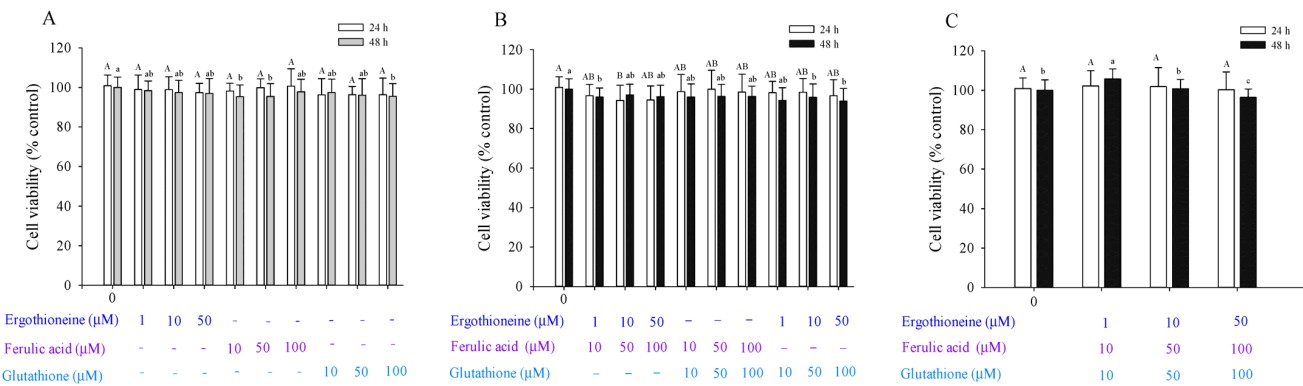

**Figure 2.** Viability of Hs68 cells after 24 h and 48 h in the presence of a single antioxidant (**A**), combination of two antioxidants (**B**), and combination of three antioxidants (**C**), namely ergothioneine, ferulic acid, and glutathione. Each value is expressed as the mean ± SD ($n \geq 3$). Different letters indicate significant differences among treatments ($p < 0.05$).

When the dose is 200 mJ/cm$^2$, the cell viability is less than 80%. UVB irradiated the cells for 24 h and 48 h, resulting in a significant decrease in cell viability by 75.55% and 74.66% (Figure 3). UVB irradiation at 200 mJ/cm$^2$ was previously shown to reduce the cell viability of human dermal fibroblasts significantly in a dose-dependent manner [34]. Based on these results, 150 mJ/cm$^2$ and an incubation time of 24 h were used for the experiments in this study. After UVB irradiation, fibroblasts were treated with various concentrations of single or combined antioxidants in 1% serum medium; this serum concentration was selected due to results of pre-study experiments, which showed that 10% serum in the medium affected cell viability to a greater degree than the single or combined antioxidants.

After UVB irradiation at 150 mJ/cm$^2$ (Figure 4), for a single antioxidant, cell viability after 24 h of ERG treatment at 1–50 μM was 88.90–86.32%; after 24 h of FA treatment at 10–100 μM, it was 98.66–95.62%; after 24 h of GSH treatment at 10–100 μM, it was 89.21–93.06% (Figure 4A). For 48 h, the effect of a single antioxidant on cell viability did not significantly differ from that of UVB irradiation of control cells (79.80%). For two an-

tioxidants, cell viability after 24 h was as follows: for the E + F group, 94.12–95.47%; for the F + G group, 86.88–91.59%; for the E + G group, 91.13–93.31% (Figure 4B). Considering the cell viability after 48 h, only the cell viability of the F50 + G50 group was higher than that of the UVB-irradiated control cells. However, the viability of the other eight groups was lower than that of the UVB-irradiated control cells. Considering the three-antioxidant combination, the cell viabilities were 91.22–94.73% after 24 h (Figure 4C). After 48 h, the cell viability after adding a single antioxidant was not significantly different from the viability of UVB-irradiated control cells. In summary, whether administered alone or in combination, antioxidants can increase cell viability after UVB irradiation.

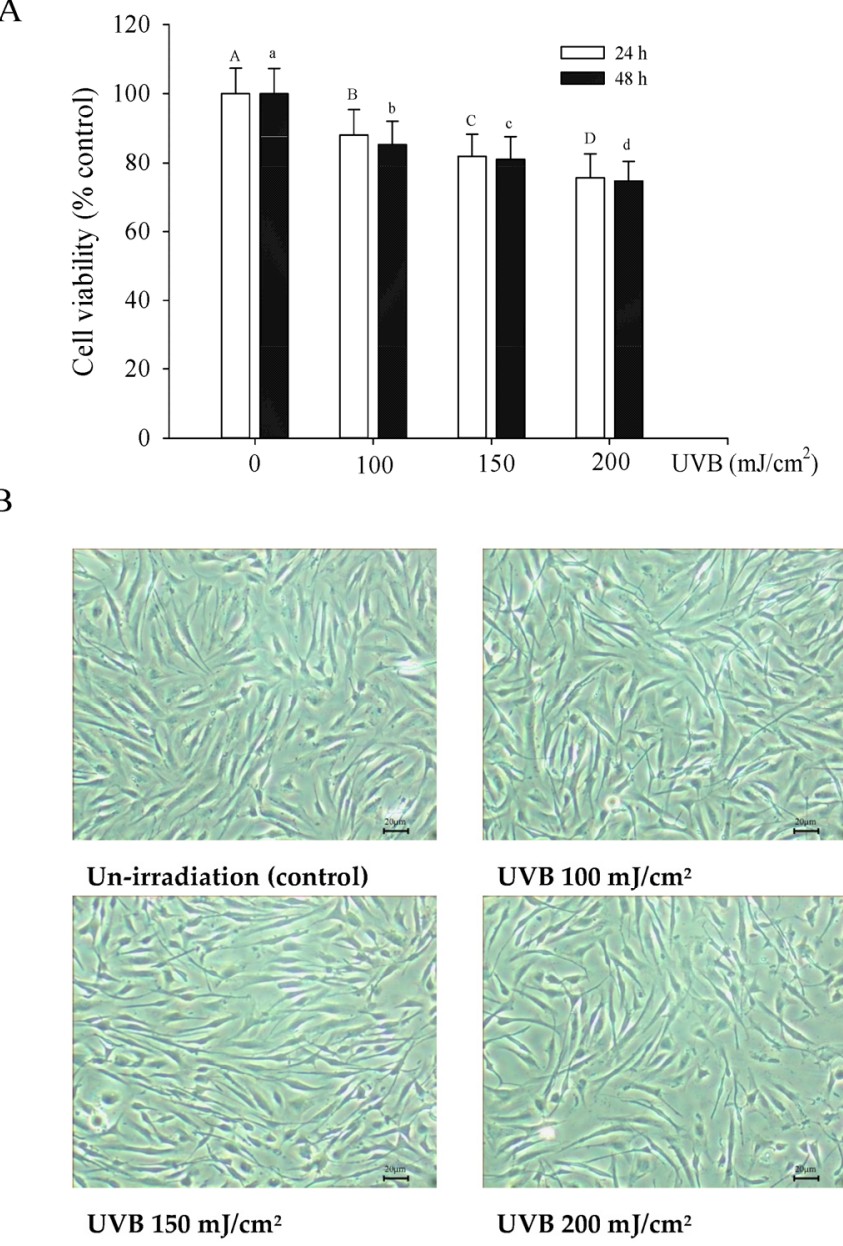

**Figure 3.** Viability of Hs68 cells for 24 h or 48 h after exposure to UVB at different irradiation dosages (**A**). Cell image of Hs68 cells for 24 h after exposure to UVB at different irradiation dosages (**B**). Each value is expressed as the mean ± SD ($n \geq 3$). Different letters indicate significant differences among treatments ($p < 0.05$).

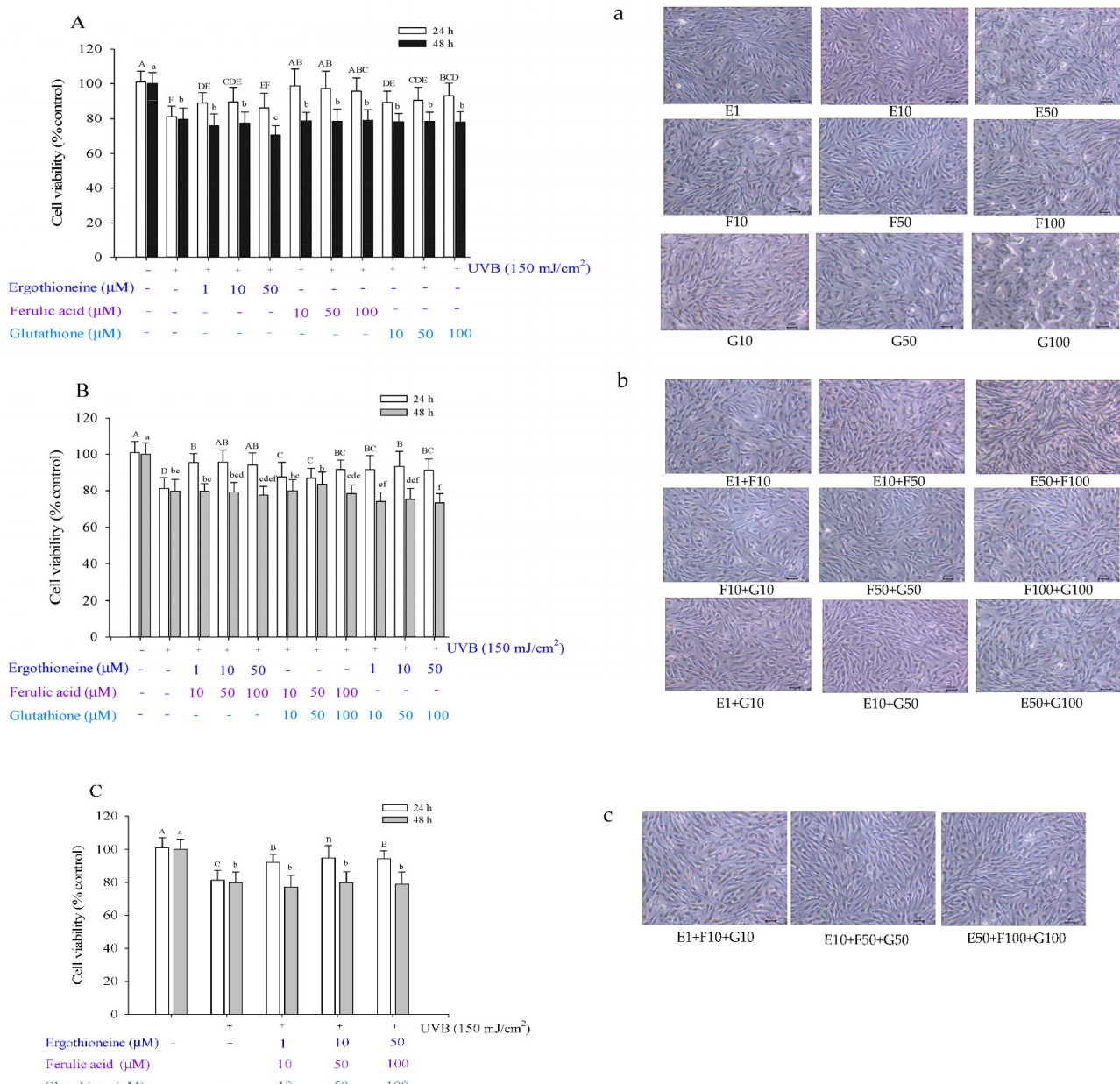

**Figure 4.** Viability of Hs68 cells after UVB irradiation for 24 or 48 h in the presence of a single (**A**), combination of two (**B**), or combination of three types of antioxidants (**C**), namely ergothioneine, ferulic acid, and glutathione. Effects of ergothioneine, ferulic acid, and glutathione on the cell image of Hs68 cells after UVB irradiation at 150 mJ/cm$^2$ for 24 h, in the presence of a single (**a**), combination of two (**b**), or combination of three types of antioxidants (**c**). Each value is expressed as the mean $\pm$ SD ($n \geq 3$). Different letters indicate significant differences among treatments ($p < 0.05$).

### 3.2. Detection of ROS Production

Previous studies have shown that UVB irradiation can induce fibroblasts to generate ROS, which is related to oxidative stress and the photodamage of skin fibroblasts [26]. ERG reduces the generation of ROS caused by either UVA or UVB irradiation of skin cells [14]. However, ROS production significantly increased in fibroblasts after UVB irradiation, with values of 1.44 (% control) and 1 (% control) for irradiated cells at 150 mJ/cm$^2$ and unirradiated control cells, respectively (Figure 5). Considering the addition of a single antioxidant, FA showed more inhibition of ROS production than either ERG or GSH, with inhibition from 112.19% to 151.63% as its concentration increased (Figure 5A). Considering

the addition of two mixed antioxidants, E + F, F + G, and E + G, inhibition effects were also seen in a dose-dependent manner (Figure 5B). In particular, E50 + F100 and F100 + G100 showed the highest inhibition effects of 135.12% and 139.00%, respectively. Considering the addition of three mixed antioxidants, inhibition ranged from 107.70% to 117.87% as concentration increased (Figure 5C). Overall, ERG, FA, and GSH were able to inhibit the ROS production that was caused by the UVB irradiation of Hs68 cells. Worthy of note are the two mixed antioxidants, which conferred greater inhibition than the three mixed antioxidants.

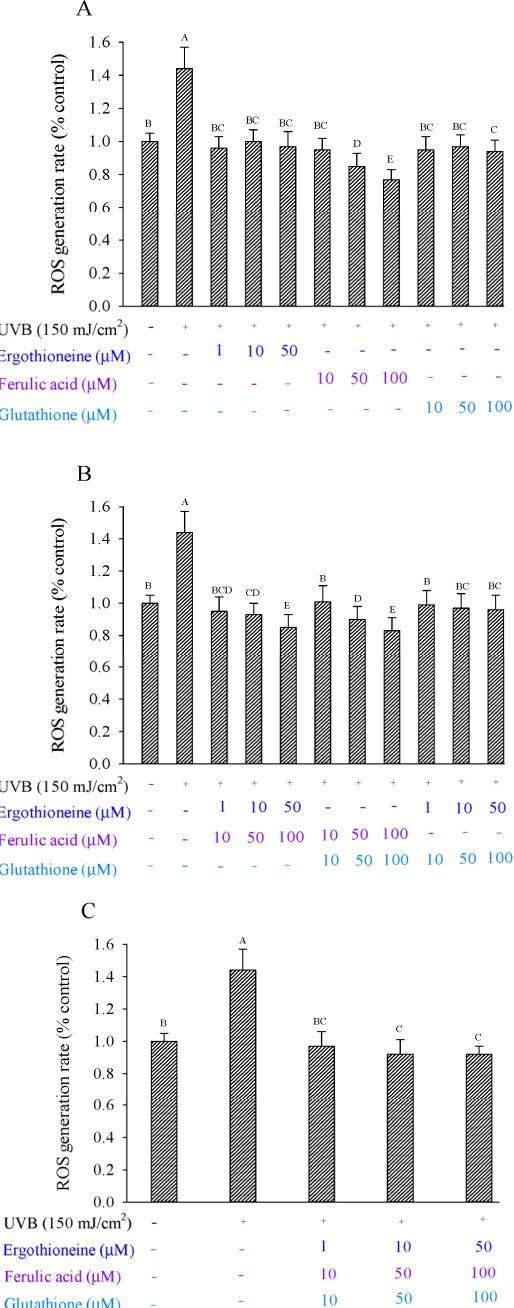

**Figure 5.** ROS production in Hs68 cells induced by UVB irradiation in the presence of a single (**A**), combination of two (**B**), or combination of three types of antioxidants (**C**), namely ergothioneine, ferulic acid, and glutathione. Each value is expressed as the mean $\pm$ SD ($n \geq 3$). Different letters indicate significant differences among treatments ($p < 0.05$).

### 3.3. Total MMP-1 Production and IL-1 Alpha

The amount of MMP-1 protein in UVB-irradiated cells was threefold greater than that in non-irradiated cells [26]. When fibroblasts were exposed to 100 mJ/cm$^2$ or 150 mJ/cm$^2$ UVB radiation, MMP-1 production was 1.98 and 2.10 ng/mL, respectively (Figure 6A), and production increased approximately twofold compared to controls (0.95 ng/mL). After 150 mJ/cm$^2$ UVB irradiation, the cells were treated with single or combined antioxidants in 1% serum medium. Regarding single antioxidants, ERG showed inhibition ranging from 74.87% to 90.03%; FA showed inhibition ranging from 42.70% to 71.36%; and GSH showed inhibition ranging from 22.15% to 68.35% (Figure 6B). ERG exhibited a greater inhibitory effect on MMP-1 production than ferulic acid or glutathione, especially at 1 μM ergothioneine. ERG had been previously shown to inhibit MMP-1 production after the UV irradiation of human dermal fibroblasts [12]. FA had been previously shown to inhibit MMP-1 expression in CCD-986sk cells after UVB irradiation [21]. Regarding the two mixed antioxidants, the F100.0 + G100.0 groups showed the greatest inhibitory effect at 82.24% (Figure 6C). Regarding the three mixed antioxidants, the E50.0 + F100.0 + G100.0 groups showed the greatest inhibitory effect, at 30.40% (Figure 6D). Overall, adding either a single antioxidant or two antioxidants were effective in reducing MMP-1 production after UVB irradiation. UV irradiation exposure results in the induction of fibroblast-derived proinflammatory cytokines. However, IL-1 alpha was not detected in the Hs68 cells after 150 mJ/cm$^2$ UVB irradiation [32].

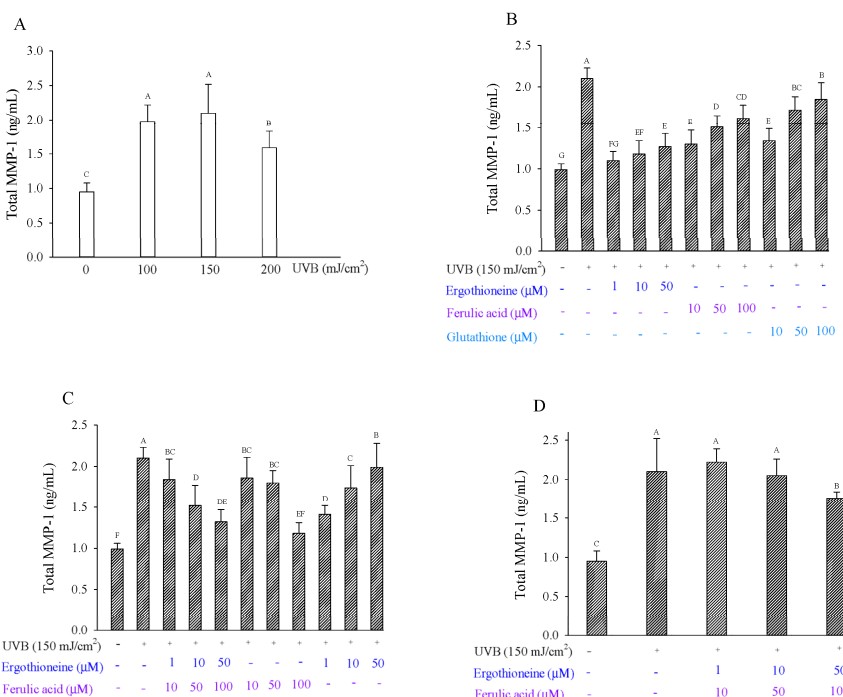

**Figure 6.** Total MMP-1 production by Hs68 cells after exposure to UVB at various irradiation dosages (**A**). Total MMP-1 production by Hs68 cells induced by UVB irradiation in the presence of a single (**B**), combination of two (**C**), or combination of three types of antioxidants (**D**), namely ergothioneine, ferulic acid, and glutathione. Each value is expressed as the mean ± SD ($n \geq 3$). Different letters indicate significant differences among treatments ($p < 0.05$).

### 3.4. Quantitative Determination of Type I Collagen Secretion

Many studies have shown that fibroblasts in the dermis reduce the production of type I procollagen after UVB irradiation [16,35]. FA has been shown to inhibit MMP-1 and MMP-9 expression in human dermal fibroblast lines as well as induce procollagen synthesis [14]. In our previous study, high-dose UVB irradiation could cause irreversible

DNA damage and apoptosis in skin fibroblasts, and after 300 mJ/cm$^2$ UVB irradiation, no type I procollagen was detected in the fibroblasts. The protective effect was evaluated by incubating Hs68 cells irradiated at 150 mJ/cm$^2$ with different extract concentrations for 24 h.

For single antioxidants in the fibroblasts, ERG showed a healing effect, increasing the percentage of viable cells from 57.57% to 77.36%, while FA showed a healing effect, with the percentage of viable cells increasing from 9.92% to 46.02% (Figure 7A). Regarding GSH, only low concentrations showed a healing effect, with 41.06% of cells remaining viable. ERG showed a greater restorative effect on type 1 procollagen production than either FA or GSH. Considering the two mixed antioxidants in fibroblasts, the E50 + F100 and F100 + G100 groups showed the greatest protective effect, with 53.91% and 41.15% viable cells, respectively (Figure 7B). When considering the three mixed antioxidants (E50 + F100 + G100) added to fibroblasts, the inhibitory effect on MMP was 2.68% (Figure 7C). Given these results, the addition of antioxidants that combine ERG with FA or FA with GSH would allow for the restoration of type I procollagen production.

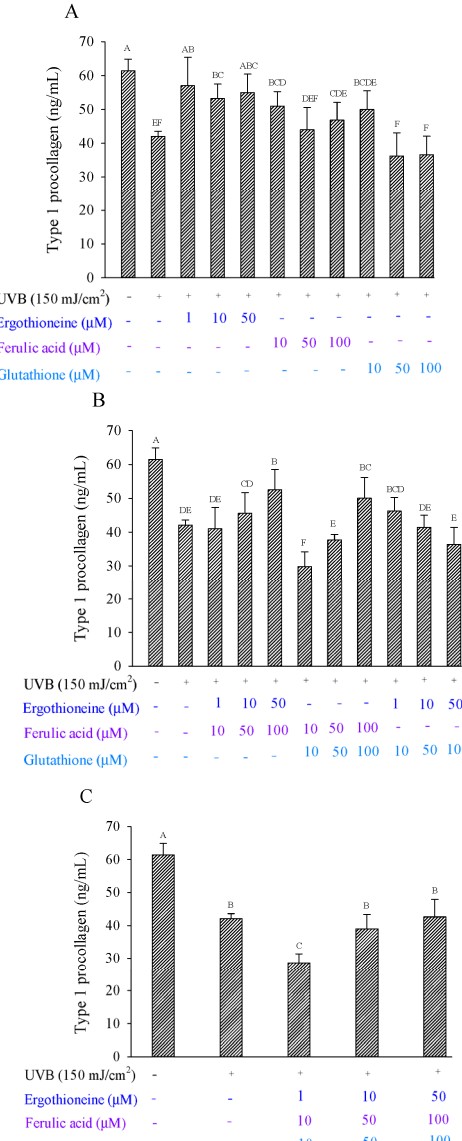

**Figure 7.** Type 1 procollagen production of Hs68 cells induced by UVB irradiation in the presence of a single (**A**), combination of two (**B**), or combination of three types of antioxidants (**C**), namely ergothioneine, ferulic acid, and glutathione. Each value is expressed as the mean $\pm$ SD ($n \geq 3$). Different letters indicate significant differences among treatments ($p < 0.05$).

Table 2 summarizes the data of this study; all of the data suggest that the higher the value, the better the light protection ability. Under the use of a single antioxidant, various concentrations of ergothioneine and ferulic acid have the best preventive effect (calculated based on the number indicated). Under the use of combined two antioxidant groups, E50 + F100 and F100 + G100 have the best preventive effect. Finally, when three kinds of antioxidants are combined, the protection effect is poor.

**Table 2.** The total includes ROS, total MMP-1, and Type 1 procollagen analysis data of three types of antioxidants in the prevention of UVB radiation-induced photoaging in human skin fibroblasts.

| Concentration (μM) | ROS(Inhibitory, %) [a] | Total MMP-1 (Inhibitory, %) [b] | Type 1 Procollagen (Recovery, %) [c] |
|---|---|---|---|
| Single antioxidant | | | |
| E 1 | 109.45 * | 90.03 ***** | 77.36 **** |
| E 10 | 100.56 * | 82.14 ***** | 57.57 *** |
| E 50 | 106.07 * | 74.87 **** | 66.72 **** |
| F 10 | 112.19 ** | 71.36 **** | 46.02 *** |
| F 50 | 133.55 **** | 53.02 *** | 9.92 * |
| F 100 | 151.63 ***** | 42.70 *** | 24.61 ** |
| G 10 | 111.84 ** | 68.35 **** | 41.06 *** |
| G 50 | 105.74 * | 34.26 ** | −30.19 |
| G 100 | 112.82 ** | 22.15 ** | −28.15 |
| Combined two antioxidants | | | |
| E 1 + F 10 | 112.23 ** | 23.14 ** | −5.50 |
| E 10 + F 50 | 115.76 ** | 51.74 *** | 18.05 * |
| E 50 + F 100 | 135.12 **** | 69.94 **** | 53.91 *** |
| F 10 + G 10 | 98.37 | 21.07 ** | −63.59 |
| F 50 + G 50 | 122.26 *** | 26.44 ** | −22.82 |
| F 100 + G 100 | 139.00 **** | 82.24 ***** | 41.15 *** |
| E 1 + G 10 | 101.95 * | 61.68 **** | 21.67 ** |
| E 10 + G 50 | 107.38 * | 32.09 ** | −3.16 |
| E 50 + G 100 | 108.14 * | 9.66 * | −29.21 |
| Combined three antioxidants | | | |
| E 1 + F 10 + G 10 | 107.70 * | −10.94 | −69.42 |
| E 10 + F 50 + G 50 | 119.12 ** | 4.34 * | −15.83 |
| E 50 + F 100 + G 100 | 117.87 ** | 30.40 ** | 2.68 * |

[a] ROS range, * 100–110; ** 111–120; *** 121–130; **** 131–140; ***** 141–150. [b] Total MMP-1 range, * 0–20; ** 21–40; *** 41–60; **** 61–80; ***** 81–100. [c] Type 1 procollagen range, * 0–20; ** 21–40; *** 41–60; **** 61–80.

## 4. Conclusions

UVB irradiation can cause serious skin problems such as DNA damage, cancer, and/or skin aging. Free radicals can destroy fibroblasts, leading to a decrease in collagen content in human skin. Over time, it has been shown that ERG is able to protect skin cells from UV irradiation as well as increase the production of type I procollagen. Furthermore, FA has been shown to protect skin from UV irradiation in some studies. The first part of our objective was to investigate the anti-photoaging activity of three antioxidants, namely ERG, FA, and GSH after the UVB irradiation of human skin fibroblasts. Our results showed that one, two, or three antioxidants can increase cell viability after UVB irradiation of the fibroblasts. ERG, FA, and GSH significantly decreased ROS production that was caused by the UVB irradiation of fibroblasts, and two mixed antioxidants were the most effective, especially the E50 + F100 and F100 + G100 groups. Adding a single antioxidant or two antioxidants was effective at reducing MMP-1 production after UVB irradiation. In particular, lower concentration of ERG showed a greater inhibition of MMP-1 performance than FA, GSH, or the combined groups. For single antioxidants, ERG showed a greater healing effect than FA and GSH. For two mixed antioxidants in fibroblasts, the E50 + F100 and F100 + G100 groups showed the greatest restorative effect on type 1 procollagen.

**Author Contributions:** Data curation, G.J.T. and C.-Y.L.; conceptualization, S.-Y.T.; methodology, S.-Y.L. and J.-L.M.; writing—original draft preparation, S.-Y.L.; writing—review and editing, S.-Y.T. All authors have read and agreed to the published version of the manuscript.

**Funding:** The authors are grateful to the Ministry of Science and Technology (MOST) and Asia University (ASIA) under the contract No.: MOST 108-2320-B-468 -001 -MY3 and ASIA-109-CMUH-07.

**Institutional Review Board Statement:** Not applicable.

**Informed Consent Statement:** Not applicable.

**Data Availability Statement:** Not applicable.

**Conflicts of Interest:** The authors declare no conflict of interest.

**Abbreviations**

| | |
|---|---|
| AP-1 | Activator protein-1 |
| DMEM | Dulbecco's Modified Eagle's medium |
| ERG | Ergothioneine |
| FA | Ferulic acid |
| GSH | Glutathione |
| MMP-1 | Matrix metalloproteinase-1 |
| MTT | [3-(4,5-dimethyl -2-thiazolyl)-2,5-diphenyl-2H-tetrazo lium bromide |
| NF-kB | Nuclear factor-Kb |
| PBS | Phosphate-buffered saline |
| ROS | Reactive oxygen species |
| TGF-$\beta$1 | Transforming growth factor-$\beta$1 |
| UV | Ultraviolet |

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
