# Peer review of "Comparison of Single and Combined Use of Ergothioneine, Ferulic Acid, and Glutathione as Antioxidants for the Prevention of Ultraviolet B Radiation-Induced Photoaging Damage in Human Skin Fibroblasts"

_processes, doi:10.3390/pr9071204_

Round 1
Reviewer 1 Report
Comments to the authors:
In the paper, Tsay et al. investigated the potential of Ergothioneine, Ferulic Acid, and Glutathione as antioxidants for the prevention of UVB-induced damage in Hs68 cells. Overall, the work is well organized.
Some comments below:
Major points:
- The UVB only killed 26% cells which make the assay window too small. Could similar/more significant protective effect be observed when enhanced irradiation is used?
- The Figure1 could be removed and used as TOC. It is just a summary of the work.
- It’s very confusing to use different letters indicating significant differences among treatments. Better to use asterisks and indicate the significance level in figure legends.
- Microscope images should display clear bars. The current one like the one in Figure3/4 etc is too difficult to recognize.
- Why higher concentration of protectant showed weaker effect? Its weird dose-dependent effect (Figure 6B).
Author Response
Point 1: In the paper, Tsay et al. investigated the potential of Ergothioneine, Ferulic Acid, and Glutathione as antioxidants for the prevention of UVB-induced damage in Hs68 cells. Overall, the work is well organized.
Some comments below:
The UVB only killed 26% cells which make the assay window too small. Could similar/more significant protective effect be observed when enhanced irradiation is used?
Response 1: First, thank you for reviewing our paper. We have made the appropriate corrections. The UVB radiation dose must not be too strong. When the dose is too high, the cells will all die first, and the protective effect of the sample cannot be tested.
Point 2: The Figure1 could be removed and used as TOC. It is just a summary of the work. It’s very confusing to use different letters indicating significant differences among treatments. Better to use asterisks and indicate the significance level in figure legends.
Response 2: We have rewritten in the current text in Figure 1.
Point 3: Microscope images should display clear bars. The current one like the one in Figure3/4 etc is too difficult to recognize.
Response 3: We have rewritten in the current text in Figure3/4.
Point 4: Why higher concentration of protectant showed weaker effect? Its weird dose-dependent effect (Figure 6B).
Response 4: When the cells are not irradiated with UVB, a high concentration of the sample will affect the cell survival rate due to the osmotic pressure. Due to the injury to the cells after irradiation, and the addition of high-concentration samples, the effect is not obvious.

Reviewer 2 Report
Comments to paper
Comparison of Single and Combined Use of Ergothioneine, Ferulic Acid, and Glutathione as Antioxidants for the Prevention of Ultraviolet B Radiation-Induced Photoaging Damage in Human Skin Fibroblasts
Interesting work about the effect of three compounds in cell lines, simultaneously, but need some explanations to become a better work
Line 76: ERG is not indicated where humans can get it from, food? in line 87, the origin of ferulic acid is explained (from cereals and …). Is it an endogenous compound like glutathione?
Line 100: the effect of UVB is explained and what about UVA and UVC?
Line 113: «UV radiation causes» UV radiation or UVB?
Line 140: Only E1 was used in the combination with F10??? It must be a mistake because there are 3 identical concentrations. The same for E1+G10
The results in the figures are not according to the concentrations described in material and methods.
Why these concentrations were selected to prepare the mixtures?
In the legend of Fig3 P is referred but in material and methods it was named alpha. It should be the same letter.
In Figure 3 the cells are shown but nothing is commented about these images.
Shouldn’t the study about the effect of UVB light come before the study of the effect of compounds preventing the bad effects of UVB? That is Figure 3 should be figure 2 and after that, the study of prevention of the antioxidant compounds should be seen. As a matter of fact, it is supposed that these compounds counteract the effect of the UV light.
Which is the difference between Figure 2 and Figure 4?
Conclusion should not have discussion but only the main findings (Some researchers have described ERG and GSH found in edible mushrooms, as important compounds in skin care products). The sentences about where to find the compounds should be indicated in the introduction or during the discussion.
Author Response
Point 1: Interesting work about the effect of three compounds in cell lines, simultaneously, but need some explanations to become a better work
Response 1: Thank you for reviewing our paper. According to your comments, we have made the appropriate corrections.
Point 2: Line 76: ERG is not indicated where humans can get it from, food? in line 87, the origin of ferulic acid is explained (from cereals and …). Is it an endogenous compound like glutathione?
Response 2: We have rewritten the description of the introduction in the text.
Point 3:
Line 100: the effect of UVB is explained and what about UVA and UVC?
Line 113: «UV radiation causes» UV radiation or UVB?
Line 140: Only E1 was used in the combination with F10??? It must be a mistake because there are 3 identical concentrations. The same for E1+G10
Response 3: We have rewritten the description of the introduction in the text.
Point 4:The results in the figures are not according to the concentrations described in material and methods.
Response 4: According to your comments, we have made the appropriate corrections in the figures.
Point 5:Why these concentrations were selected to prepare the mixtures?
Response 5: According to the cell test, 80% of the cell survival rate is the selected concentration.
Point 6:In the legend of Fig3 P is referred but in material and methods it was named alpha. It should be the same letter. In Figure 3 the cells are shown but nothing is commented about these images.
Response 6: We have rewritten the description of the methods and results in the text.
Point 7:Shouldn’t the study about the effect of UVB light come before the study of the effect of compounds preventing the bad effects of UVB? That is Figure 3 should be figure 2 and after that, the study of prevention of the antioxidant compounds should be seen. As a matter of fact, it is supposed that these compounds counteract the effect of the UV light.
Response 7: Thank you for reviewing our paper. According to your comments, we have made the appropriate corrections. Cell test requires sample toxicity test before UVB damage to cells
Point8 :Which is the difference between Figure 2 and Figure 4?
Response 8: Figure 2 shows the cytotoxicity test of the sample itself. Figure 4 shows the toxicity test of the sample to the cell after the cell is irradiated with UVB.
Point9 :Conclusion should not have discussion but only the main findings (Some researchers have described ERG and GSH found in edible mushrooms, as important compounds in skin care products). The sentences about where to find the compounds should be indicated in the introduction or during the discussion.
Response 9: According to your comments, we have made the appropriate corrections.

Reviewer 3 Report
The authors present the use of three antioxidants, alone or in combination, for the prevention of skin damage in an in vitro model of human fibroblasts under UVB radiation exposure. The protective effect of these antioxidants has been previously described in the literature, but the novelty of this work is the combination of them. The performance of the antioxidants in three concentrations has been evaluated by cell viability assays, ROS production, IL-1a, total MMP-1 levels, Procollagen 1 alpha 1.
There are many writing issues that need to be addressed. It is unclear which combination (or antioxidant alone) is the most appropriate to follow up on these studies. It would be great to add a table summarizing and putting all the findings together.
Line 126: Remove “of”.
The description of the methodology is clear; however, the result section does not follow the same structure and it is confusing. For example, 2.4 describes methods for cell viability and calculation of healing effects, but in results, cell viability and healing effects are separated, being the latest in the procollagen section. Please clarify this and be consistent throughout all the text.
What was the highest cell passage number?
Line 212: replace “was” by “were”
What do you mean by subsamples? How do you split/get them?
Mention which are the controls for each experiment.
Line 224: are those independent experiments? If yes, please, mention “independent experiments” in the text.
Line 226: standard error or standard deviation?
Line 242: Check FA concentrations.
Line 244-246: That phrase is confusing there and could lead the readers to think the authors have isolated the compound. Please rephrase.
Lines 260-262: Please rephrase.
Lines 283-284: Please rephrase.
Lines 286-288: Please rephrase.
Line 286: Be consistent, always use “h”.
Line 297-298: 1.44- and 1-fold? Be consistent in the text and the figures with the % values of ROS production.
“IL-1 alpha was not detected in the Hs68 cells after 150 mJ/cm2 UVB irradiation.” Please discuss this. Did you expect this? What does indicate? Is it in agreement with the literature?
Line 346: Please start the sentence with a capital letter.
lines 361-363: Greatest but very low, almost negligible.
Line 364: any two antioxidants?
Conclusions are not well-written. It looks like a discussion and a summary of the results. At this point, it is not clear for this reviewer what is the best way to prevent the UVB radiation damage on the skin fibroblasts. Discussion is also poor, most of the findings did not have their explanation or the plausible explanation.
According to instructions for authors, the order in the manuscript should be Introduction, Results, Discussion, Materials and Methods, Conclusions (optional).
Figures
Replace plots in bars for plots with dots.
Increase the size of the scale bars on the images.
Figure 1: The flow diagram doesn’t show the correct order. Also, I would not call this a “flow diagram”.
In all figure legends, the symbol is wrong (P > 0.05). Please replace by “<” (less than).
Figure 6D: Isn’t the “y-axis” label wrong? I think it should be Total MMP-1 …
Author Response
Point 1: The authors present the use of three antioxidants, alone or in combination, for the prevention of skin damage in an in vitro model of human fibroblasts under UVB radiation exposure. The protective effect of these antioxidants has been previously described in the literature, but the novelty of this work is the combination of them. The performance of the antioxidants in three concentrations has been evaluated by cell viability assays, ROS production, IL-1a, total MMP-1 levels, Procollagen 1 alpha 1.
Response 1: Thank you for reviewing our paper. According to your comments, we have made the appropriate corrections.
Point 2: There are many writing issues that need to be addressed. It is unclear which combination (or antioxidant alone) is the most appropriate to follow up on these studies. It would be great to add a table summarizing and putting all the findings together.
Response 2: We have rewritten in the current text. Please see Table 1 and Table 2.
Point 3: Line 126: Remove “of”.
Response 3: We have rewritten the description of the introduction in the text.
Point 4: The description of the methodology is clear; however, the result section does not follow the same structure and it is confusing. For example, 2.4 describes methods for cell viability and calculation of healing effects, but in results, cell viability and healing effects are separated, being the latest in the procollagen section. Please clarify this and be consistent throughout all the text.
Response 4: We have rewritten the description of the methods for cell viability in the text. Please see the “Materials and Methods” in the current text.
Point 5: What was the highest cell passage number?
Response 5: The growth number of purchased generation cells is n=19, and the number of cells during the experiment is n+5~n+9.
Point 6: Line 212: replace “was” by “were”
Response 6: We have rewritten the description of the methods in the text.
Point 7: What do you mean by subsamples? How do you split/get them?
Response 7: We have rewritten the description of the methods and results in the text.
Point 8: Mention which are the controls for each experiment.
Response 8: According to your comments, we have made the appropriate corrections.
Point 9: Line 224: are those independent experiments? If yes, please, mention “independent experiments” in the text.
Response 9: We have made the appropriate corrections.
Point 10: Line 226: standard error or standard deviation?
Response 10: We have made the appropriate corrections.
Point 11: Line 242: Check FA concentrations.
Response 11: We have made the appropriate corrections.
Point 12: Line 244-246: That phrase is confusing there and could lead the readers to think the authors have isolated the compound. Please rephrase.
Response 12: We have made the appropriate corrections.
Point 13:
Lines 260-262: Please rephrase.
Lines 283-284: Please rephrase.
Lines 286-288: Please rephrase.
Line 286: Be consistent, always use “h”.
Line 297-298: 1.44- and 1-fold? Be consistent in the text and the figures with the % values of ROS production.
Response 13: According to your comments, we have made the appropriate corrections.
Point 14: “IL-1 alpha was not detected in the Hs68 cells after 150 mJ/cm2 UVB irradiation.” Please discuss this. Did you expect this? What does indicate? Is it in agreement with the literature?
Response 14: We have made the appropriate corrections.
Point 15:
Line 346: Please start the sentence with a capital letter.
lines 361-363: Greatest but very low, almost negligible.
Line 364: any two antioxidants?
Response 15: We have made the appropriate corrections.
Point 16: Conclusions are not well-written. It looks like a discussion and a summary of the results. At this point, it is not clear for this reviewer what is the best way to prevent the UVB radiation damage on the skin fibroblasts. Discussion is also poor, most of the findings did not have their explanation or the plausible explanation.
Response 16: We have made the appropriate corrections.
Point 17: According to instructions for authors, the order in the manuscript should be Introduction, Results, Discussion, Materials and Methods, Conclusions (optional)
Response 17: We have made the appropriate corrections.
Point 18: Figures
Figures Replace plots in bars for plots with dots.
Increase the size of the scale bars on the images.
Figure 1: The flow diagram doesn’t show the correct order. Also, I would not call this a “flow diagram”.
In all figure legends, the symbol is wrong (P > 0.05). Please replace by “<” (less than).
Figure 6D: Isn’t the “y-axis” label wrong? I think it should be Total MMP-1 …
Response 18: We have rewritten in the current text.
